# A big-data approach to understanding metabolic rate and response to obesity in laboratory mice

June K Corrigan[1], Deepti Ramachandran[1], Yuchen He[1], Colin J Palmer[1], Michael J Jurczak[2†], Rui Chen[3], Bingshan Li[3], Randall H Friedline[4], Jason K Kim[4,5], Jon J Ramsey[6], Louise Lantier[3], Owen P McGuinness[3], Mouse Metabolic Phenotyping Center Energy Balance Working Group, Alexander S Banks[1]*

[1]Division of Endocrinology, Diabetes and Metabolism, Beth Israel Deaconess Medical Center and Harvard Medical School, Boston, United States; [2]Division of Endocrinology, Yale University School of Medicine, New Haven, United States; [3]Department of Molecular Physiology and Biophysics, Vanderbilt University School of Medicine, Nashville, United States; [4]Program in Molecular Medicine, University of Massachusetts Medical School, Worcester, United States; [5]Division of Endocrinology, Metabolism, and Diabetes, Department of Medicine, University of Massachusetts Medical School, Worcester, United States; [6]Department of Molecular Biosciences, School of Veterinary Medicine, University of California, Davis, Davis, United States

*For correspondence:
asbanks@bidmc.harvard.edu

Present address: †Division of Endocrinology and Metabolism, University of Pittsburgh School of Medicine, Pittsburgh, United States

Competing interests: The authors declare that no competing interests exist.

**Abstract** Maintaining a healthy body weight requires an exquisite balance between energy intake and energy expenditure. To understand the genetic and environmental factors that contribute to the regulation of body weight, an important first step is to establish the normal range of metabolic values and primary sources contributing to variability. Energy metabolism is measured by powerful and sensitive indirect calorimetry devices. Analysis of nearly 10,000 wild-type mice from two large-scale experiments revealed that the largest variation in energy expenditure is due to body composition, ambient temperature, and institutional site of experimentation. We also analyze variation in 2329 knockout strains and establish a reference for the magnitude of metabolic changes. Based on these findings, we provide suggestions for how best to design and conduct energy balance experiments in rodents. These recommendations will move us closer to the goal of a centralized physiological repository to foster transparency, rigor and reproducibility in metabolic physiology experimentation.

## Introduction

Mice are an instructive tool for the study of human metabolism as they can mirror human physiology in their responses to age-related and diet-induced obesity, and their physiological compensations to resist weight loss (*Speakman et al., 2007*). The study of genetic and environmental factors influencing energy balance using laboratory animals has been advanced by the introduction of indirect calorimetry systems (*Even and Nadkarni, 2012*). Indirect calorimeters measure metabolic rates using gas sensors to capture rates of change in $O_2$ consumption and $CO_2$ production within an open flow system. Physical activity is monitored by recording infrared beam breaks or using electromagnetic receivers. Food intake is measured using sensitive mass balances. Weight gain results when food intake outpaces metabolic rate. Inversely, when food intake falls below metabolic rate, weight loss

**eLife digest** Maintaining a healthy weight requires the body to balance energy intake and expenditure. The body converts food to energy through a process called energy metabolism. Genetic and environmental factors can affect energy metabolism and energy balance contributing to conditions like obesity.

To better understand metabolism, scientists often study mice in laboratories, but mice from different laboratories appear to convert food to energy at different rates. This makes it hard to determine what is 'normal' for mouse metabolism. These discrepancies could be due to small differences between how mice are kept in different laboratories. For example, the temperatures of the mouse cages or how active the mice are might differ depending on the laboratory. Identifying the effects of such differences is essential, but it requires looking at data from hundreds of mice.

Corrigan et al. examined data from more than 30,000 mice at laboratories around the world to show that room temperatures and the amount of muscle and fat in a mouse's body have the biggest influence on energy balance. These two factors affected the metabolism of both typical mice and mice with mutations that affect their energy balance.

These results suggest that it is important for scientists to report factors like room temperatures, the body make-up of the mice, or the animals' activity levels in metabolism studies. This can help scientists compare results and repeat experiments, which could speed up research into mouse metabolism. Corrigan et al. also found that other unknown factors also affect mouse metabolism in different laboratories. Further studies are needed to identify these factors.

ensues. Physiological constraints limit both unrestrained weight gain and weight loss. The ability to promote durable weight loss through increases in metabolic rate or decreases in food intake is a major therapeutic goal in the context of the modern obesogenic environment.

It is important to be able to compare the physiological data across papers studying metabolism. However, comparing results from published indirect calorimetry studies has been hampered by inconsistent application of analytical techniques. In studies of obesity, metabolic rates of mice with different body compositions are sometimes subjected to inappropriate normalization attempts (*Arch et al., 2006*; *Katch, 1972*; *Kleiber, 1932*; *Tanner, 1949*). Position papers decrying this situation have been routinely produced, and just as often ignored. A turning point occurred with the publication of a strongly worded commentary accusing authors of incorrectly representing their metabolic analyses (*Butler and Kozak, 2010*). A result of the commentary was a new appreciation of the complexity involved in handling the large amount of data resulting from indirect calorimetry experiments and the development of new tools (*Mina et al., 2018*). Many groups have agreed that the optimal data treatment uses the ANCOVA, an ANOVA with body mass or body composition as a covariate (*Arch et al., 2006*; *Kaiyala, 2020*; *Kaiyala, 2014*; *Kaiyala et al., 2010*; *Kaiyala and Schwartz, 2011*; *Speakman et al., 2013*; *Tschöp et al., 2012*). However, the implementation of these recommendations has been uneven (*Fernández-Verdejo et al., 2019*). To address this problem, we developed *CalR*, which facilitates investigators uploading and analyzing their calorimetry data by automating many of the routine steps of data curation, pre-defining statistical significance cutoffs, and allowing users to automatically perform appropriate statistical tests for interaction effects (*Mina et al., 2018*). *CalR* is the first step toward standardization of indirect calorimetry analysis across different equipment platforms. However, we realized that the results provided by *CalR* included measures of statistical significance, but were lacking critical physiological context.

To establish the normal range of metabolic rate under standardized experimental conditions, we analyzed two large independent datasets. For our primary training model, we utilized a dataset from the Mouse Metabolic Phenotyping Centers (MMPCs) representing longitudinal data from four sites in the United States where groups of male C57BL/6J mice were followed for 12 weeks on either a standard low-fat diet (LFD) or an obesogenic high-fat diet (HFD). The results of our analysis from the MMPC experiment were applied to a larger secondary dataset from the International Mouse Phenotyping Consortium (IMPC), an ambitious large-scale project with metabolic data from more than 30,000 mice, seeking to determine the genetic contributions to mammalian physiology. Here, we

present the results of our analyses from these two datasets, and provide generalized recommendations to facilitate the creation of a centralized metabolic data repository.

## Results

### Factors influencing metabolic rate in mice

To understand the different components affecting mouse metabolism, cohorts of genetically identical mice were shipped to four independent US MMPCs and assessed longitudinally over 12 weeks in indirect calorimeters while on LFD or HFD. Weekly body masses were similar among institutions for mice on LFD but diverged significantly among mice on HFD (*Figure 1A and B*). Regression plots of 24 hr average energy expenditure (EE) versus total body mass showed distinct site-specific metabolic rates which could be attributed to differences in body mass for mice on LFD or HFD (*Figure 1C and D*). Mice at each site and overall had a positive association between mass and EE. This mass effect reflects the biological consequence of Newton's 2nd law of motion, whereby movement of greater mass requires more energy, and by extension larger animals require more energy for both resting metabolism and to perform work under standard conditions (*Kleiber, 1932*; *White and Seymour, 2005*). For mice on LFD, animals at each institution formed distinct slopes and intercepts indicative of location-specific differences in housing temperature, experimental apparatus, microbiome, and other factors. For animals on 4 or 11 weeks of HFD, we observed identical slopes with different intercepts (*Figure 1D*). This suggests that the relationship of EE to mass among genetically identical mice is similar despite absolute differences in EE at different locations.

### Low-fat and high-fat diets

Despite an identical genetic background, a single source colony for all mice, as well as a single source of animal diet, an unexpectedly large weight variation was observed by site among the 30 mice randomly assigned to HFD. Body masses and compositions diverged appreciably, both among institutions and within each site's colony. The mass variation was not observed for mice on LFD, suggesting strong effects driving obesity which are not encoded by genetic variation in C57BL/6 mice (*Figure 1E*). We sought to quantify the relative contribution of the likely sources for the variation in EE using a widely adopted method (*Grömping, 2006*). The $R^2$ of 72% reflects total explained variance and a good overall fit. This analysis reveals the largest source of variation in metabolic rate is body composition. Other sources include activity, time of day (photoperiod), diet and acclimation (*Figure 1F*). The institutional site of experimentation which can affect these factors accounts for 16.3% of the residual variation not explained by these biological factors (*Figure 1G*).

### Role of acclimation on indirect calorimetry data

Rodents engage in a behavioral response when challenged with a novel environment (*Archer, 1975*). While there are no firm guidelines for how long mice may need to acclimate to indirect calorimeters, age, strain and diet can impinge on acclimation time. Here we defined the acclimation period as the first 18 hr or prior to the start of the first full photoperiod in the calorimeter. Analyses of the pre-acclimation and post-acclimation dark photoperiods were performed for each site. The hourly mean values for EE, energy intake, and respiratory exchange ratio (RER) from all four sites are plotted vs time. The strong effects of entrainment to 12 hr light/dark photoperiods produce differences at all sites for all parameters measured. The effects of acclimation were highly variable, differentially affecting mice at each location. At one of the four sites, EE increased following acclimation (*Figure 1—figure supplement 1A and B*). Similarly, energy intake was increased at only one site in non-acclimated mice (*Figure 1—figure supplement 1C and D*). The RER value, representing substrate oxidation, was significantly altered in three of four sites (*Figure 1—figure supplement 1E and F*). In this study, locomotor activity was not significantly impacted by lack of acclimation (*Figure 1—figure supplement 1G and H*). Overall, lack of acclimation adds a small but unpredictable level of noise and variation to the measurements tested.

### Role of mass and body composition in energy expenditure

How HFD feeding contributes to obesity has been a topic of detailed investigation. Both male and female C57BL/6 mice typically gain weight on a diet high in fat (*Johnston et al., 2007*; *West et al.,*

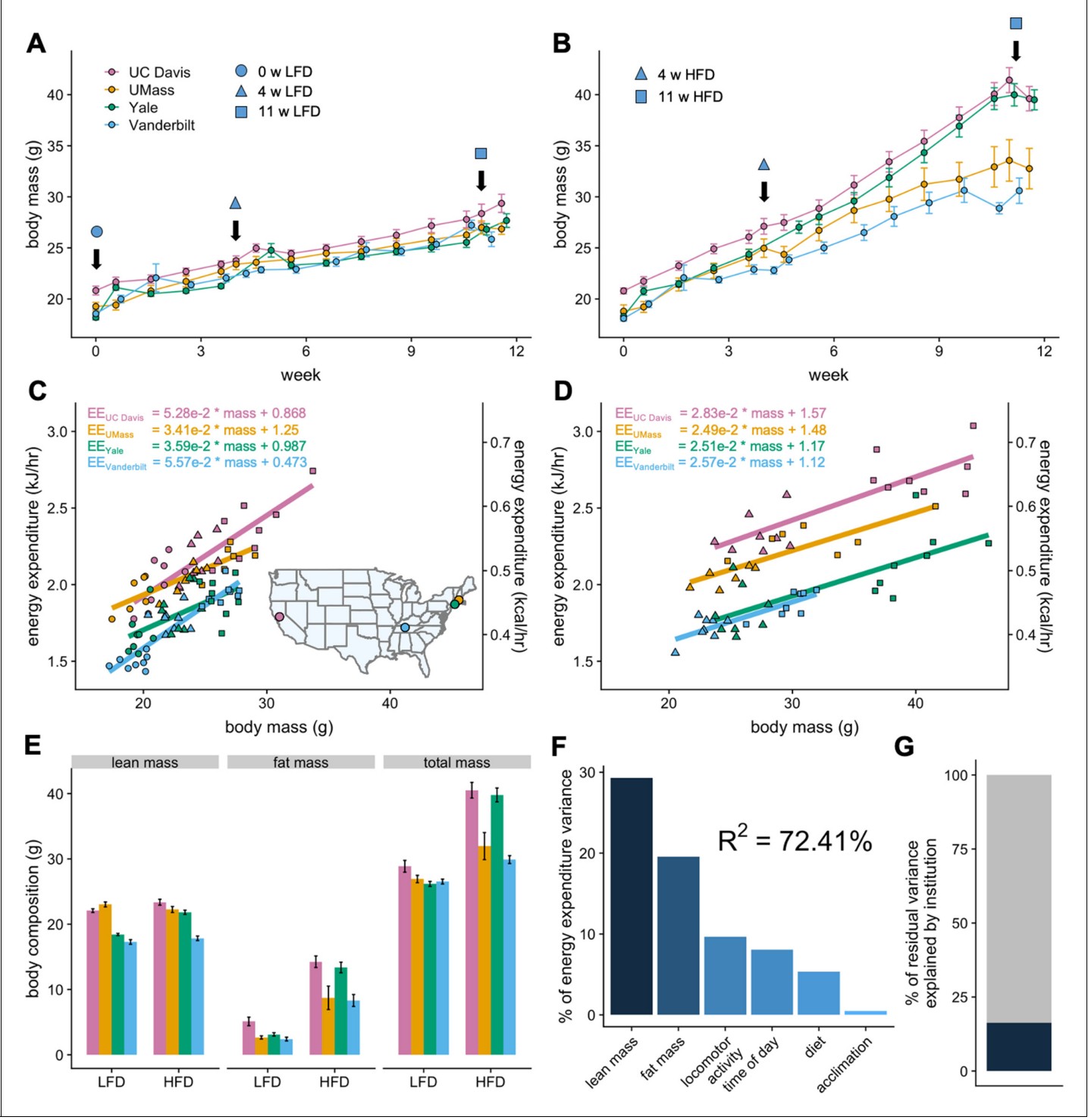

**Figure 1.** MMPC experiment: institutional variability in response to low-fat or high-fat diet. Body weights of WT C57BL/6J mice maintained on LFD (**A**) or HFD (**B**). Indirect calorimetry was performed at 0 (*o*), 4 (Δ), or 11 (□) weeks. EE rates vs total body mass are plotted by site for mice maintained on LFD (**C**) or HFD (**D**), with the slope and intercept in kJ/hr indicated for each site. Inset, geographical locations of MMPC sites. Body composition data from mice at 11 weeks on diet (**E**). The percent variation in EE explained by each of the listed factors (**F**). The otherwise unattributed institutional variation (16%) (**G**). n = 6–8 males per group. Error bars represent SEM.

The online version of this article includes the following figure supplement(s) for figure 1:

**Figure supplement 1.** MMPC experiment: effect of acclimation.

*1992*). The degree of weight gain however is highly variable and consequently there is considerable heterogeneity in the underlying mechanisms involving energy intake, metabolic rate, and absorption (*Kohsaka et al., 2007*; *Lin et al., 2000*; *Mayer and Yannoni, 1956*; *Storlien et al., 1986*; *Yang et al., 2014*). Here, we compare the calorimeter-determined energy intake and EE responses of WT male mice to HFD feeding at four different locations. After 11 weeks on LFD or HFD, as before, the EE was dependent on body mass (slopes of kJ/hr vs gram body weight) for all mice at all sites. However, HFD did not significantly alter this relationship except at one center (*Figure 2A*). We fitted the LFD mouse data at each site to a model including body mass and activity to predict expected values for mice on HFD. The difference from the expected values, or residual values from this fit show an overall decrease in the EE of HFD mice relative to the predicted EE (*Figure 2B*). We similarly examined how rates of energy intake changed with diet and found a surprising degree of variance among sites (*Figure 2C and D*). Body weight is ultimately affected by long-term differences in energy balance (energy intake minus EE). When calculating the difference between calories consumed and calories expended, as before, we observed a large variation in responses to HFD, including positive and negative changes in energy balance (*Figure 2E and F*). However, energy intake was the primary driver of energy balance. These findings serve to underscore the large site-specific effects that contribute to variability of metabolism. To evaluate the contribution of body composition to the variable metabolic rates observed between sites, we used lean mass values as the covariate for our ANCOVA analysis (*Figure 2—figure supplement 1A–1C*). In three of four sites, HFD altered the dependency of EE on lean mass. At one site, HFD altered the dependency of energy intake on lean mass. We also explicitly calculated the relative contribution of fat mass to EE using multiple linear regression and found an unexpectedly large range of fat mass contributions varying from +33% to −19% (*Figure 2—figure supplement 1D*).

When we examined the time course of the calorimetry data for all the centers, they demonstrated the characteristic circadian patterns of EE, activity and RER in animals on LFD during the 3-day period (*Figure 2G, L and H*). Moreover, on HFD, EE was increased, and RER was decreased. Total energy intake and cumulative intake were increased by HFD, yet energy balance was unaltered by diet reflecting a neutral energy balance and weight stability (*Figure 2K*) while in the calorimeter. We also plotted change in energy balance vs weight change between the start and end of the first calorimetry experiment at week 0 (*Figure 2—figure supplement 2A*). The normal circadian patterns seen in these mice, as well as the positive linear relationship between their energy balance and mass balance (i.e. animals in positive energy balance tend to gain weight) speak to the quality of the acquired data.

## Application to large-scale dataset. A role for location, temperature, mass and sex

To understand the relevance of the initial analysis, we applied this approach to the IMPC dataset (*Rozman et al., 2018*). We similarly plotted the change in energy balance vs weight change between the start and end of the IMPC calorimetry experiments. The data appear to be of high quality as indicated by the expected positive linear relationship. Furthermore, mice in positive energy balance showed a higher RER consistent with carbohydrate oxidation, and mice with weight loss exhibit a lower RER consistent with oxidizing endogenous fat stores (*Figure 2—figure supplement 2B*).

The IMPC data demonstrate a pronounced mass effect, representing that bigger male and female animals expend more energy under standard housing conditions. The IMPC data include results of indirect calorimetry performed on 32,748 distinct animals including 9358 WT C57BL/6N mice at 10–11 weeks of age. 69% of these mice were male. The mean body mass of female mice is lower than that for male mice (at 21.7 g and 27.4 g respectively; *Figure 3A*). In this analysis, there is a small but significant difference in the slope of the relationship between EE and body mass in WT male and female mice (*Figure 3B* and *Figure 3—source data 1*).

We next sought to understand whether it is appropriate to compare mice of different weights. To do this, we examined whether the slope of the relationship between EE and body mass changes when comparing mice over a large range of masses (*Figure 3—figure supplement 1A*). We grouped all male and female WT mice into quartiles by body mass (small, 14.00–20.75 g; medium, 20.75–27.50 g; large, 27.50–34.25 g; largest, 34.25–41.00 g), and found that there were significant differences in the slope of the relationship between EE and body mass among the four groups. Somewhat surprisingly, largest mice are not significantly different from any of the other groups, which may be

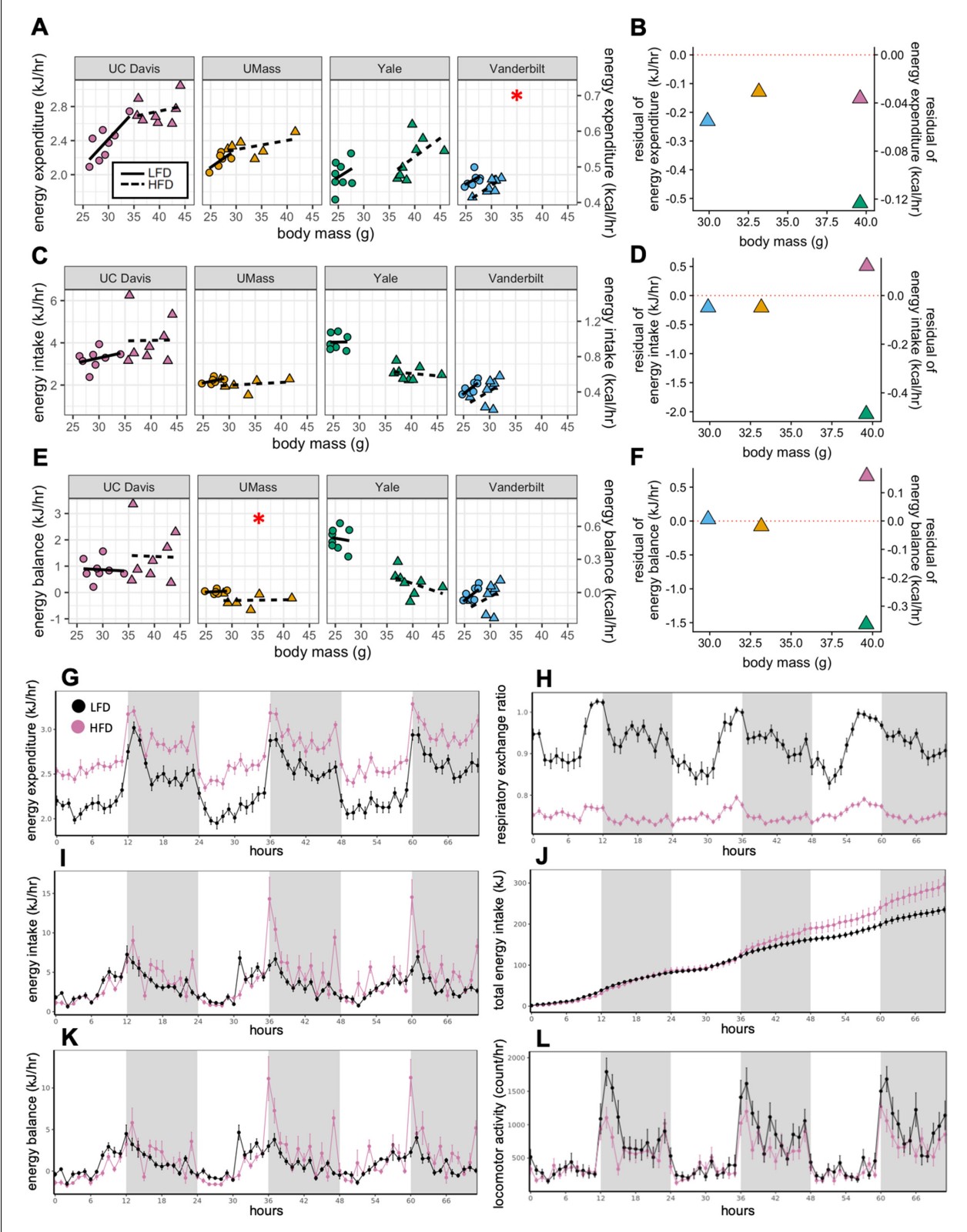

**Figure 2.** MMPC experiment: institutional variability in metabolic rate. Regression plots for each of the MMPC sites vs mass (left). A regression model fitted to LFD mice for site, mass, and locomotor activity was established; the deviation from the predicted values is shown as residuals (right). EE (**A–B**). Energy intake (**C–D**). Energy balance (**E–F**). Time-dependent plots for the UC Davis site after 11 weeks on HFD including EE (**G**), RER (**H**), hourly energy

*Figure 2 continued on next page*

*Figure 2 continued*

intake (**I**), cumulative energy intake (**J**), energy balance (**K**), and locomotor activity (**L**). Values are hourly means (**G–L**). *, p<0.05. n = 6–8 males per group. Error bars represent SEM. Shaded regions represent the dark photoperiod from 18:00 to 6:00.

The online version of this article includes the following figure supplement(s) for figure 2:

**Figure supplement 1.** MMPC experiment: energy balance using lean mass as a covariate.
**Figure supplement 2.** Body weight, energy balance and RER.

attributed to the fact that there are the fewest mice in the group with the largest masses. Our main finding shows the 'large' mice have a significantly shallower slope in the relationship between body mass and EE compared to both small and medium mice. This finding was similar to the relationship observed in the MMPC cohort on LFD and HFD (*Figure 2A*). This is due to the differential contribution of lean and fat mass to EE which are not equally accumulated in mice with greater body weight. Indeed, when examining only lean body mass as the covariate, there were no significant differences between any of the groups by ANCOVA (*Figure 3—figure supplement 1B*). This finding suggests that when comparing mice with more than 20% difference in body mass, that body composition data must be added to an ANCOVA model so as to not produce spuriously significant results.

## Variability in WT female mouse metabolic rate

There is no indication that metabolic rate of female mice is more variable than that of male mice; a long-held assumption in metabolic research, despite recent contradictory evidence (*Prendergast et al., 2014*). Studies on mice of both sexes were performed at seven IMPC institutions; three institutions examined only male mice (*Figure 3—figure supplement 2A and B*). To determine the variability in metabolic rate, we fitted EE to the available data for each institution and both sexes. The $R^2$ value of this fit represents the quality of the fit, with larger numbers representing a better fit, with lower unexplained variability. The $R^2$ values varied considerably by site, likely due to six sites reporting locomotor activity values and only three sites reporting accurate ambient temperature data, while all sites reported body mass. Despite the great variation between sites, $R^2$ values were similar between males and females at six of the sites (+/- 7.3% difference). The remaining site had a female $R^2$ of less than 0.001, a poor fit, due to a small number of mice with little variation in mass, the sole predictor in this model, and therefore not an instructive example. The Canadian site reported mass, temperature and activity, and despite having the third smallest sample size had the best fit with $R^2_{female}$ = 0.507 and $R^2_{male}$ = 0.468. Combining the data from both sexes at this site further improved the fit to 0.581 suggesting only minor differences in metabolic rate among female and male young, chow-fed WT mice. Clearly, the more accurate the covariate information reported, the better the explanatory value of the model. As the sample sizes of the two sexes are unequal in these sites (2889 females and 6469 males), we also examined the distribution of variance in male and female mice with a modified quantile-quantile (Q-Q) plot. Here, the slope of the blue and pink lines represents the theoretical standard deviation (SD) for the EE of each sex. Parallel lines indicate similar variability; while points not falling on the line represent variation from the normal distribution. The greater slope for male mice is indicative of greater variability in the data. Overall, this representation suggests that female mice have a qualitatively better fit and lower overall variation (*Figure 3—figure supplement 2C*).

## Institutional variability

Due to the significant differences in EE observed due to sex among all WT IMPC mice, for subsequent analysis of control and experimental mice, we present results only for male mice henceforth, but we find qualitatively similar metabolic results for WT male and female mice in the IMPC dataset when analyzed separately. The 10 sites represented in the IMPC data each reported between 240 and 1367 male WT mice (*Figure 3C*). We observed more than a three-fold difference in metabolic rate slopes (1.18–5.32 × $10^{-2}$ kJ/hr vs gram body mass) between sites despite standardized experimental protocols on mice of similar ages and similar diets (*Figure 3D* and *Figure 3—source data 1*). To further understand these differences, we examined other factors which influence metabolic rate including mass, temperature, season, and locomotor activity. The ambient temperature for mouse experiments varied by site. A range of ambient temperatures were reported

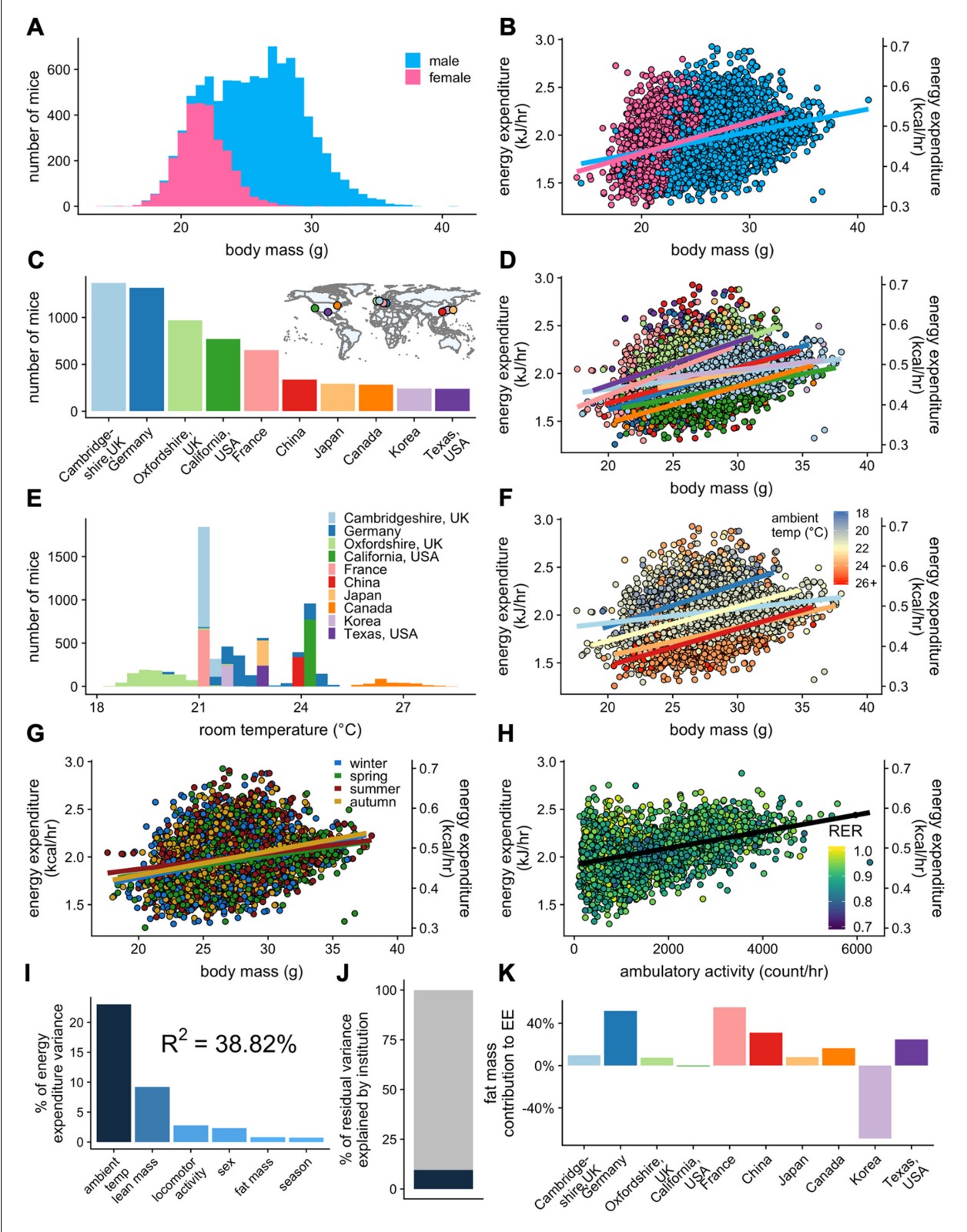

**Figure 3.** IMPC experiment: institutional variability of metabolic rate. Distribution of body weights for male and female WT C57BL/6N mice from the IMPC database version 10.0 (A). Relationship of EE vs total body mass for female and male WT mice (B). The numbers of male WT mice examined at each of the 10 IMPC sites (C). Inset: geographical locations of IMPC sites. Relationship of EE vs total body mass for male WT mice at each of the 10 IMPC sites (D). Reported ambient temperatures for the experiments performed at each site (E). Relationship of EE vs total body mass in WT males for

*Figure 3 continued on next page*

*Figure 3 continued*

temperature (F) and season (G). Relationship of EE vs ambulatory activity in WT mice (n = 3886 males) (H). Unaccounted institutional contribution here is approximately 10% (J). The percent variation in EE explained by each of the listed factors (I). Percent contribution of fat mass to EE by site in WT males (K). n = 6469 males and 2889 females unless otherwise specified.

The online version of this article includes the following source data and figure supplement(s) for figure 3:

**Source data 1.** Pairwise comparisons of slopes in regression plots of EE vs total mass.
**Figure supplement 1.** IMPC experiment: body weight and metabolic rate.
**Figure supplement 2.** Variation in energy expenditure for WT female and male mice.
**Figure supplement 3.** Irregularities in the IMPC dataset.

for experiments performed by the sites in Canada, Germany, and Oxfordshire, UK. However, most sites report one single nominal temperature (e.g. 21.0°C) which likely does not accurately reflect daily variation in room temperature, nor the actual cage temperature, and may be a major source of unexplained variation (*Figure 3E*). WT mice housed at low temperatures had the highest EE, and conversely mice housed at warmer temperatures had the lowest EE (*Figure 3F*). We next examined the effect that time of year (i.e. season) might be having on metabolic rate. In the wild, mammals have seasonal differences in metabolic rate. There is evidence that despite the climate-controlled atmosphere of a vivarium, laboratory mice could similarly detect changes in season (*Drickamer, 1977*). All sites contributing data (Canada, China, France, Germany, Japan, Korea, UK, and USA) are located in the Northern Hemisphere and experience seasonal differences (*Figure 3C*, inset). Within these data, there were no significant differences in EE by season, suggesting that seasonal variation was not a significant contributor to experimental variability (*Figure 3G*). Differences in locomotor activity can contribute to whole-body EE, but typically have a negligible effect at non-thermoneutral temperatures (*Virtue et al., 2012*). We find a strong positive correlation between ambulatory locomotor activity and EE in the six sites reporting movement data (Canada, Oxfordshire UK, Japan, France, Korea, and Cambridgeshire UK) (*Figure 3H*). We examined the relative importance of these effects to predict rates of EE (*Figure 3I*). As with the MMPC dataset, lean mass is one of the top predictors of variability, though temperature (not included in the MMPC data) contributes the highest to explaining EE rate. Unlike the MMPC dataset, more than 60% of the variation in the IMPC data was unaccounted for by examining only these reported variables, suggesting that our model is as yet incomplete. Only 9.6% of the residual variance can be attributed to unaccounted for institutional differences (*Figure 3J*). We also examined the contribution of fat mass to overall EE and found a highly variable contribution from +55% to −69% among sites (*Figure 3K*). The variability is likely due to their low adiposity as these animals are on chow diets. Overall, our analysis of 9358 WT mice identifies housing temperature, body composition, locomotor activity, and sex as contributing variables to EE.

To test the applicability of our analysis to the IMPC dataset, we chose the data from the three sites that reported both body composition and ambient temperature to within 0.1°C: Toronto Centre for Phenogenomics (Canada), Helmhotlz Munchen (Germany), and MRC Harwell (Oxfordshire, UK). We examined the contribution of total body mass to EE. Within this subset of 1996 male and 1153 female mice we re-examined sex as a mass dependent covariate. We find no significant difference between the EE rates of male and female mice (*Figure 4A*). Using total body mass as a covariate, we observe nearly identical slopes for EE vs mass at each institution (*Figure 4B*). When temperature effects were examined at these sites, mice maintained at warmer temperatures (22–29°C) had significantly lower EE rates than mice maintained at colder temperatures (*Figure 4C* and *Figure 3—source data 1*). We find an improved $R^2$ of 67% for this dataset of 3149 mice (*Figure 4D*). This example accounts for only three sites, but now has accurate ambient temperature, mass, sex and season for all animals. These data do not account for locomotor activity. This greater fit has improved predictive powers over sites without accurate temperature recording. Here, a mere 0.078% of residual variation is explained by institution (*Figure 4E*).

## Variability in KO phenotypes

After observing the large inter-site variation for WT mice, we next examined whether phenotypic differences due to genetic modulation could be observed reproducibly at multiple sites. In the IMPC

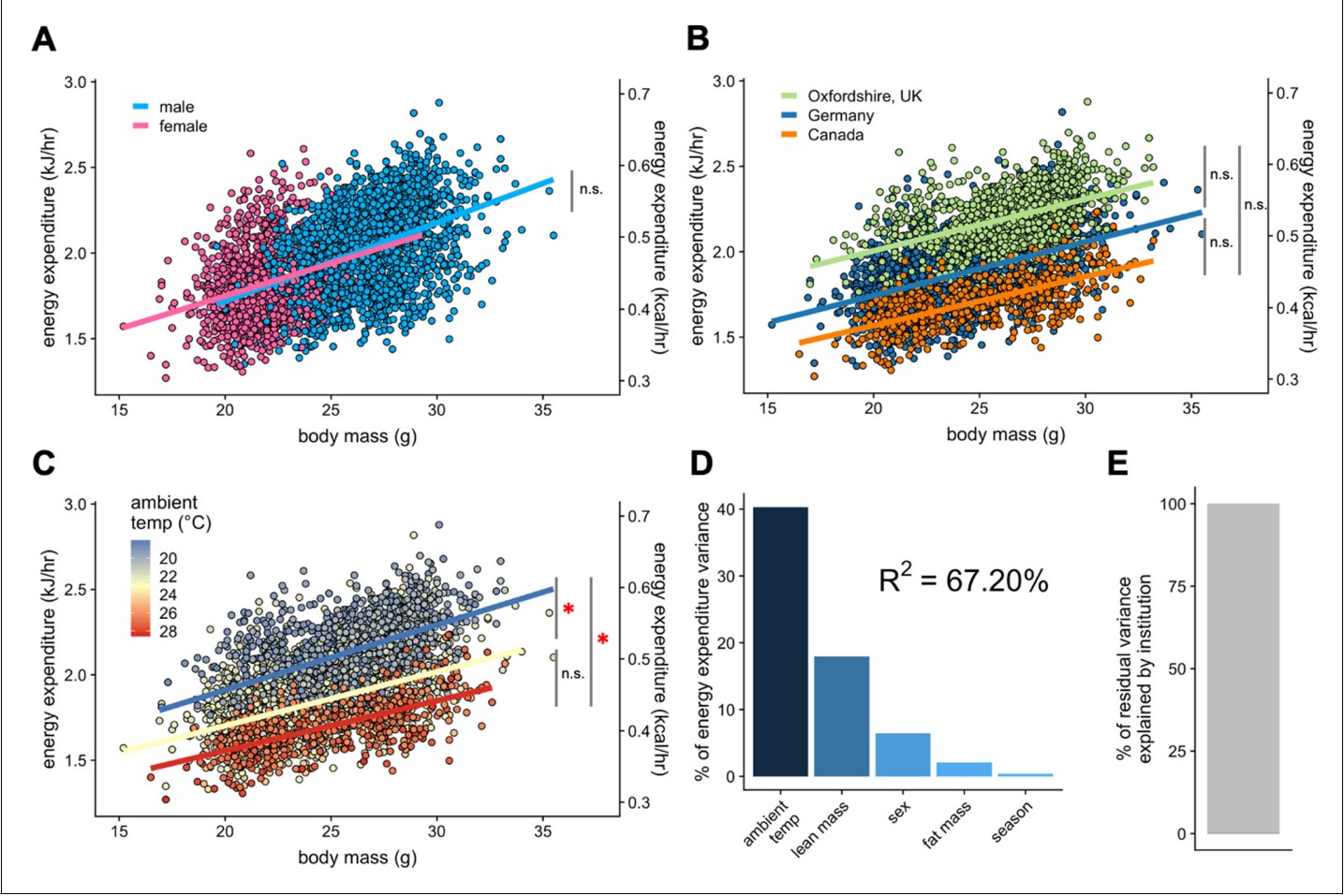

**Figure 4.** IMPC experiment: Comparison of three sites by sex and temperature. Regression plots of EE vs total body mass for male and female WT mice (**A**). Institutional effects for male and female mice on regression plots of EE vs total mass (**B**). Temperature effects for male and female mice on regression plots of EE vs total mass by ambient temperature (blue, cold regression line <22°C; yellow, mid 22–26°C; red, hot >26°C) (**C**). The percent variation in EE explained by each of the listed factors (**D**). Unaccounted institutional variation is less than 1% (**E**). *, p<0.05. n = 1996 males and 1,153 females.

dataset, there were six strains examined in at least four locations, *Ap4e1*$^{-/-}$, *Dbn1*$^{+/-}$, *Dnase1l2*$^{-/-}$, *Nxn*$^{+/-}$, and *Prkab1*$^{-/-}$ (*AMPKβ1*$^{-/-}$), and *Rnf10*$^{-/-}$. We also compared results for the *Cdkal1*$^{-/-}$ gene from two IMPC sites with our results (Massachusetts, USA). We plotted EE vs body mass for male mice at each of the sites to examine phenotypic variability (*Figure 5A*). This representation plots each individual mouse at each site but does not convey the difference from the local WT populations and thus is challenging to make direct comparisons. To provide context for the magnitude of these phenotypes, mass, activity (if available), and temperature were fit to a multiple linear regression model for WT male mice. We calculated the deviation from the model at each site and plotted these residual values (*Figure 5B*). Four of these strains have not previously been published and thus the expected phenotype is not known. The only strain with consistent phenotypic findings was *Cdkal1*$^{-/-}$, where EE values similar to WT were observed for mice at all three locations tested (*Palmer et al., 2017*). Unidirectional phenotypes were observed for four strains. Significantly increased EE was observed in *Ap4e1*$^{-/-}$, *Dbn1*$^{+/-}$, and *Rnf10*$^{-/-}$ mice at four of nine sites, three of seven sites, and two of four sites, respectively. *Nxn*$^{+/-}$ mice had significantly lower EE at three of eight sites. Bidirectional changes were observed for the remaining two strains. *Dnase1l2*$^{-/-}$ mice were similar to WT controls at six sites, with two sites showing significantly altered EE, with one higher and one lower. *Prkab1*$^{-/-}$ mice were similar to controls at five sites, consistent with results from an independently generated strain of AMPKβ1 deficiency which found no significant EE phenotype (*Dzamko et al., 2010*). Yet at

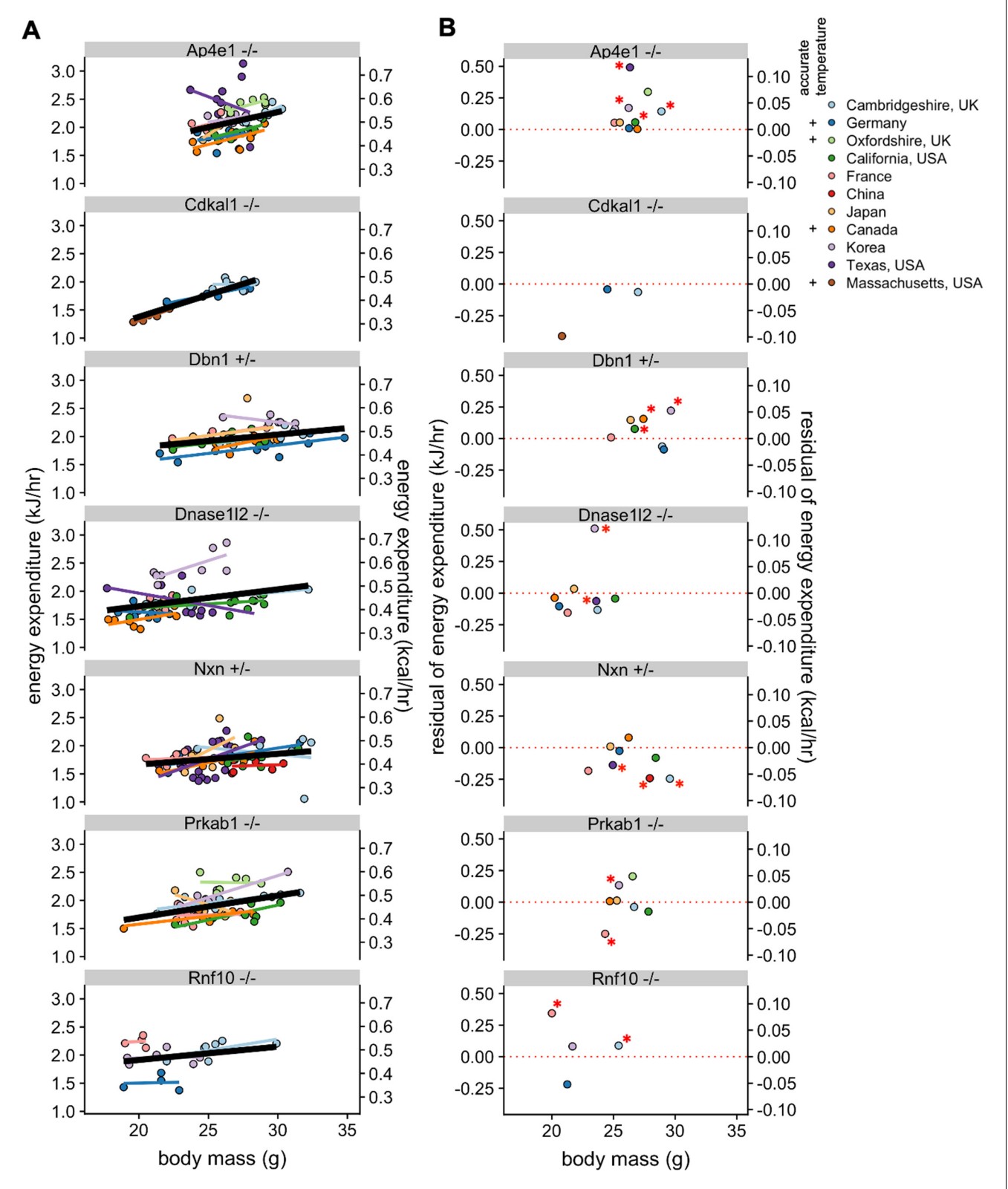

**Figure 5.** Reproducibility of KO phenotypes. Plots of EE vs mass for strains that were studied at in least four sites and also for Cdkal1, studied at two IMPC sites and in Massachusetts, USA, with overall best fit regression line shown in bold. (**A**). Mean residual values from a regression model showing the difference from the controls at the specific site (**B**). n = 4–21 males per group. *, p<0.05.

two sites *Prkab1*[-/-] mice were statistically different, higher and lower than controls at one site each. Consistently, the largest residual values were from sites that did not report accurate temperature values or locomotor activity, pointing toward incomplete modeling from these locations. These results illustrate the large phenotypic variability among mice with identical genetic perturbations observed at different locations.

## Model application: genes with known effect on body weight

Our analysis helps to quantify the magnitude by which site, temperature, mass, and sex contribute to metabolic rate in WT mice. However, genetic variations can also impact metabolic rate. One of the goals of the IMPC is to use mouse models to understand the contribution of genetic factors to phenotypic variation. The IMPC dataset contains the metabolic analysis of 2329 gene KO strains. Using the multiple linear regression model fitted from WT male mice, we examined the phenotype of all KO strains. To find the strains of KO mice with the greatest impact on metabolic rate we fitted the effects of mass and ambient temperature. After fitting the mean EE of each KO strain to the model, we plotted the unexplained metabolic effect, or EE residuals. The residuals are normally distributed around 0 for male mice (*Figure 6A*) suggesting the model is appropriately fitted (*Figure 6B*). The results of this analysis are included in *Figure 6—source data 1*. To validate this data treatment, we plotted the IMPC data for both EE vs total body mass (*Figure 6C*) and residual EE values from the regression model vs total body mass (*Figure 6D*). Using the IMPC data, we plotted the results for 10 genotypes previously shown to affect body weight in mice. These strains affect weight by different mechanisms, including by increasing food intake, limiting metabolic rate, and affecting energy absorption. Food intake drives obesity in strains deficient for melanocortin 2 receptor accessory protein 2, *Mrap2*[-/-] (*Asai et al., 2013*), carboxypeptidase E, *Cpe*[-/-] (*Alsters et al., 2015*; *Naggert et al., 1995*), proprotein convertase 1, *Pcsk1*[+/-] (*O'Rahilly et al., 1995*; *Zhu et al., 2002*), or growth differentiation factor 15, *Gdf15*[-/-] (*Tsai et al., 2013*). Low metabolic rate drives obesity in mice lacking growth hormone, *Gh*[-/-] (*Meyer et al., 2004*). Relatedly, at standard room temperatures, mice lacking the thermogenic uncoupling protein 1 have reduced metabolic rate, but obesity is not observed in *Ucp1*[-/-] mice until housing under thermoneutral conditions (*Enerbäck et al., 1997*). Mice lacking the biosynthetic enzyme for creatine, *Gatm*[-/-], have also have a reduced metabolic rate, similar to findings in adipocyte specific knockout mice (*Kazak et al., 2017*). We also plotted the metabolic rate of strains which promote leanness due to high metabolic rate, *Pparg*[+/-], *Fgfr4*[+/-], *Fgfr4*[-/-], and *Acer1*[-/-]. Peroxisome proliferator activated receptor-γ, *Pparg* heterozygous mice have increased metabolic rate driven by increased locomotor activity, controlled by the central nervous system (*Lu et al., 2011*). Elevated metabolic rate in mice with fibroblast growth factor receptor 4, *Fgfr4* knockout and antisense inhibition have elevated metabolic rate likely due to increased levels of FGF19 and FGF21 (*Ge et al., 2014*; *Yu et al., 2013*). Alkaline ceramidase 1, *Acer1*-deficient mice have a skin barrier defect and progressive hair loss, likely accounting for compensatory increases in metabolic rate (*Liakath-Ali et al., 2016*). The uniform phenotyping standards of the IMPC allow a comprehensive comparison of relative phenotypic magnitude. Institutional effects on EE can make the true magnitude of effects between control and experimental strains difficult to discern in regression plots of EE vs mass (*Figure 6C*). Instead, when plotting against residual values, it is apparent that four of these genes have modest phenotypes residing within 1 SD of the predicted mean, *Cpe*[-/-], *Mrap2*[-/-], *Gdf15*[-/-], and *Ucp1*[-/-] (*Figure 6D*). While the modestly decreased EE may contribute to obesity in the *Cpe*[-/-], and *Mrap2*[-/-] mice, food intake and other factors are likely the primary drivers as reported. There was no suggestion of altered EE in *Gdf15*[-/-], consistent with emerging reports that Gdf15 induction is required for altered body weight phenotypes (*Coll et al., 2020*). The −1.0 SD phenotype observed in the *Pcks1*[+/-] mouse has a large impact on EE to predispose to obesity (*O'Rahilly et al., 1995*; *Zhu et al., 2002*). Both *Gh*[-/-] and *Gatm*[-/-] groups have lower body weights than control mice, yet have lower than predicted EE as significant contributors to obesity in these strains. The hypermetabolic 2 SD and 3 SD effects of *Fgfr4*[+/-], *Fgfr4*[-/-], and *Acer1*[-/-] contribute to resistance to obesity, while the effects of *Pparg* heterozygosity are modest at just over 1 SD. The EE effects have been combined with other assays including skin barrier function or HFD challenge to explain these phenotypes. In concert with comprehensive phenotyping, the IMPC data have confirmed these previously published findings and demonstrated the robustness and utility of the IMPC experiment.

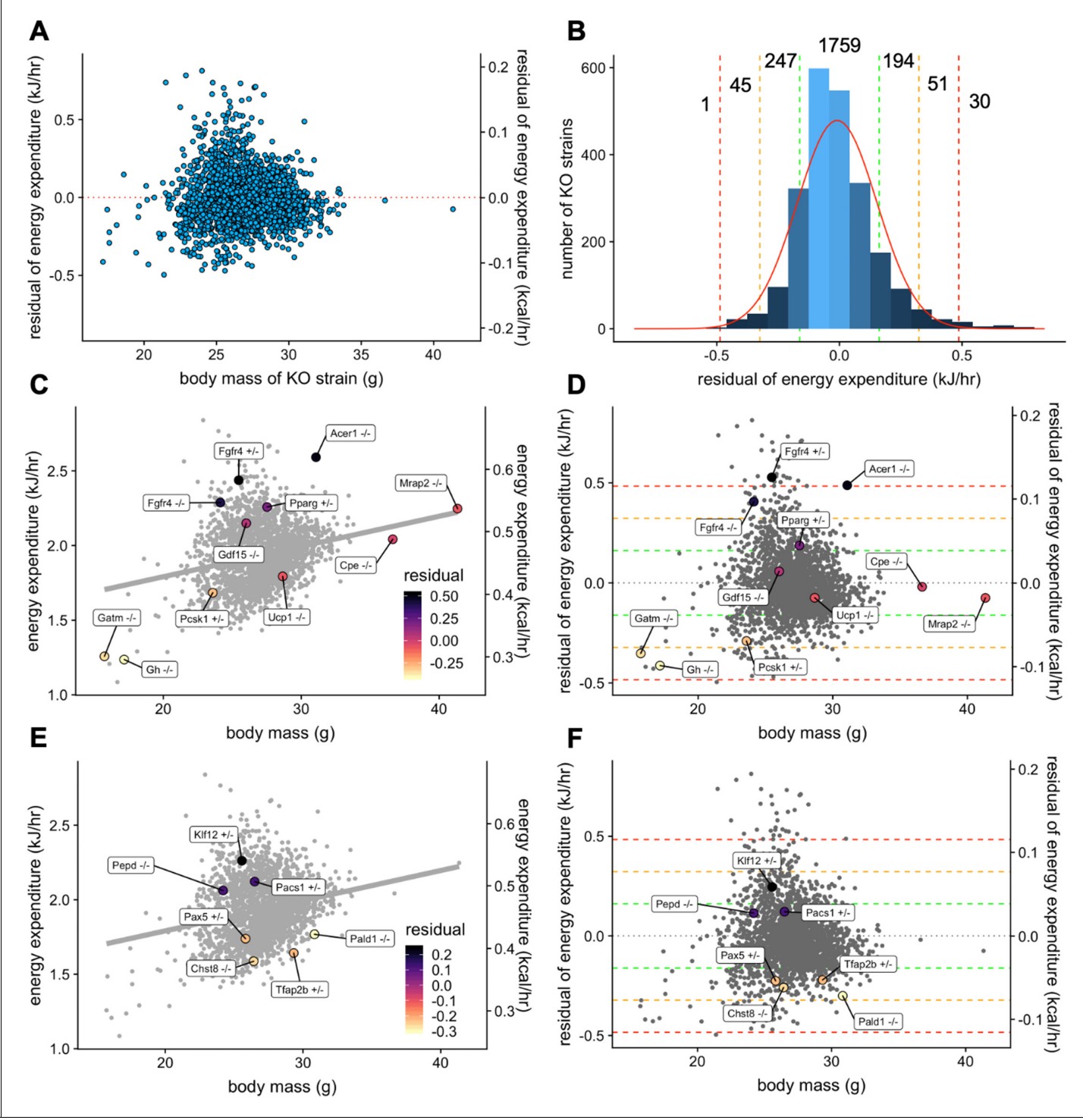

**Figure 6.** Genetic contribution to EE. The residual change in EE compared to site-appropriate controls for males of 2329 strains (**A**). Distribution of residual changes in EE for all KO strains with the numbers of KO strains reaching 1SD (green), 2SD (orange), or 3SD (red). Regression plots of EE vs mass (**C**) or residual EE vs mass for mice from mouse strains with known metabolic phenotypes (**D**). Lines showing ranges of standard deviations in EE are overlayed. Similar plots for IMPC strains with genes identified by obesity related GWAS and having phenotypes of more than 1 standard deviation (**E**, **F**). n = 23,309 males.

The online version of this article includes the following source data for figure 6:

**Source data 1.** IMPC dataset for 2329 knockout strains with residual and standard deviation values for EE.

## Model application: unknown genes

We next examined the human genetic loci linked to increased risk for obesity and body weight related traits through genome wide association studies (GWAS). The IMPC strains with a gene in a mapped obesity locus (n = 42) with phenotypes +/- 1 SD from the mean (n = 7) were plotted (*Figure 6E and F*). We found four strains with lower metabolic rate that might predict obesity susceptibility, *Pax5*$^{+/-}$ (*Melka et al., 2012*), *Chst8*$^{-/-}$ (*Tachmazidou et al., 2017*), *Tfap2b*$^{+/-}$ (*Lindgren et al., 2009*), and *Pald1*$^{-/-}$ (*Cotsapas et al., 2009*). We also found three KO strains which might protect from obesity, *Pepd*$^{-/-}$ (*Shungin et al., 2015*), *Klf12*$^{+/-}$ (*Jiang et al., 2018*), and *Pacs1*$^{+/-}$ (*Wheeler et al., 2013*). Further characterization of these strains, including exposure to HFD, may reveal insights into the phenotypic effects driving common forms of human obesity.

## Physiological context

The normal distribution of changes in EE which are not accounted for by site, mass, and temperature is relatively narrow, with 2/3 of all strains (1 SD) having modest differences of +/- 0.161 kJ/hr. To place these values into a physiological context we compared the magnitude of examples of the effects of age, voluntary exercise, adrenergic activation, and temperature on metabolism in C57BL/6 mice (*Figure 7A*). The effects of aging on basal metabolic rate, as reported by *Houtkooper et al. (2011)* were modest, with differences at 15, 55, and 94 weeks of age differing by less than 0.04 kJ/hr on average, all corresponding to less than 1 SD (*Figure 7B*). In voluntary wheel running, *O'Neal et al., 2017* report a strong (3 SD) induction of EE after one week, which decreases to a 2 SD effect at 2 and 3 weeks of exercise (*Figure 7C*). In mice housed at thermoneutrality, administration of the β3 adrenergic agonist CL316,243 produces a > 3 SD effect over 3 hr (*Figure 7D*). We also plotted metabolic rates at ambient temperatures varying from 6°C to 30°C (*Figure 7E*). Using 22°C as the comparator, increasing the temperature just 3 degrees produced a 2 SD effect. Similarly, decreasing temperature 4 degrees has a nearly 3 SD effect. Greater temperature changes produced larger effects on EE except for the transition from 28°C to 30°C which likely reflected a departure from true thermoneutrality. These plots help to provide a frame of reference for the magnitude of effects observed in both the MMPC and IMPC datasets.

# Discussion

## Small-scale data, large-scale data and data sharing

The past decade has seen a transformation in our understanding of mammalian metabolism. New genetic tools, increased commitment to large-scale experimentation, and greater sharing of data are welcome developments. Large-scale systemic experiments such as the IMPC provide an unprecedented view into the genetic pathways that control energy balance in mammals. Yet, despite information on thousands of mice being deposited annually into the IMPC, data from far greater numbers of experimental animals are produced at institutions worldwide. These smaller datasets remain siloed and unshared at great loss to the scientific community. A critical unmet need exists to create centralized data repositories on metabolism if we are to make sense of often conflicting information on the drivers of obesity. Differences in observed phenotypes may well be due to environmental variances including temperature and diet. Preclinical studies in model organisms should pave the way forward for greater understanding of the role of genetic variation, diet, exercise, macronutrient composition, age, and obesity on human metabolism. A metabolism data repository is the obvious next step given these advances. The necessary prerequisites to achieve this reality are: 1) standardization of experimentation and analysis, 2) standardization of data formats, and 3) consensus approaches to essential metadata collection. The results of experiments in mice vary by strain, age, diet, temperature, site, intervention protocols, and other factors, and there can be useful covariates in explaining variance among experiments. The absence of this information can turn an attempt at experimental replication into part of an ongoing replication crisis in biomedical research.

## Essential data to report

The goal of the present study was to understand the critical variables that would be necessary to make in vivo studies of metabolic rate broadly comparable. The largest source of variation observed in the small-scale pilot MMPC experiment was mass, including body composition data. Locomotor

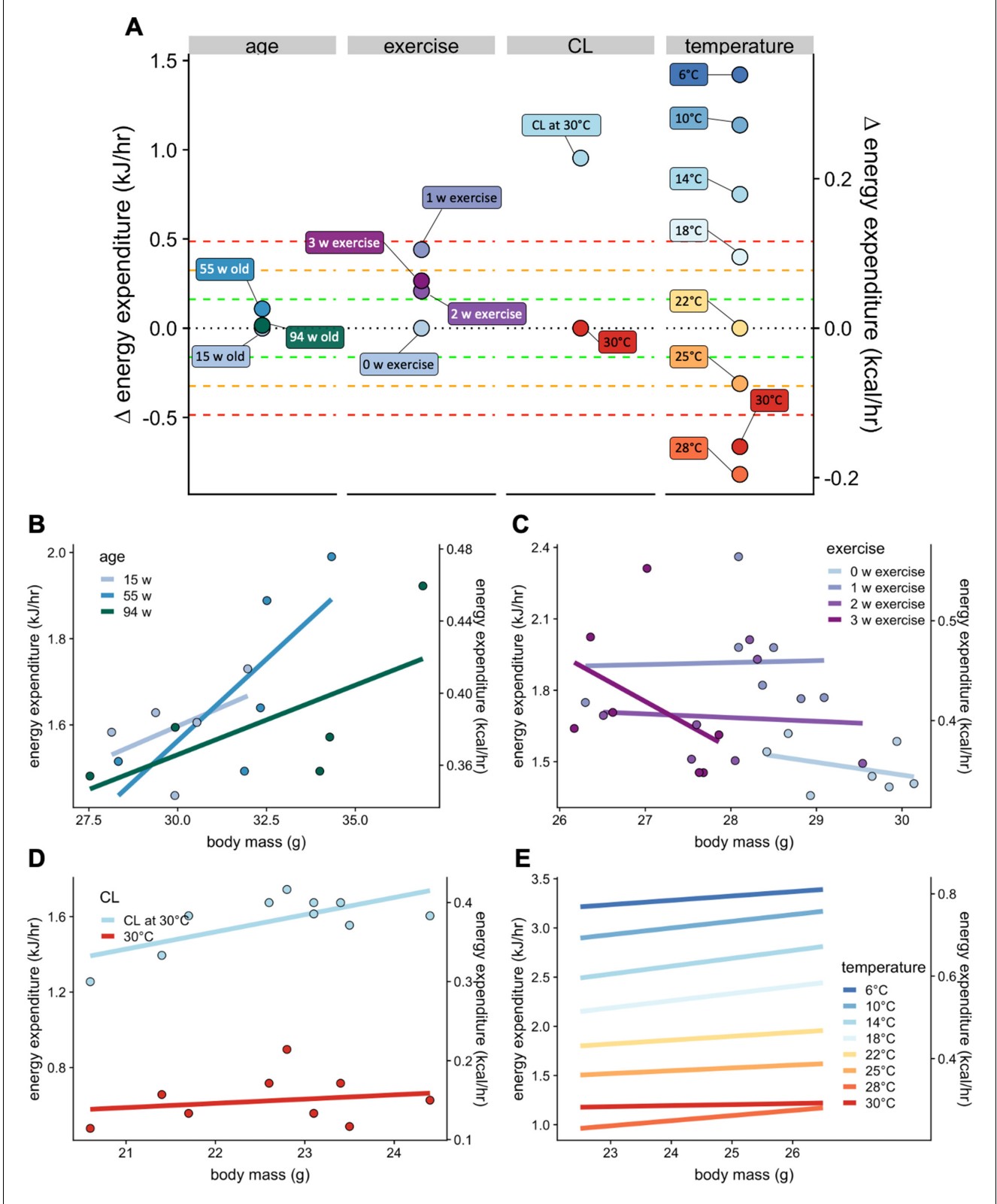

**Figure 7.** Select physiologic effects on EE. Changes in EE for physiological challenges including age, voluntary wheel running exercise, β3 adrenergic agonism, warm and cold temperature compared to relative genetic changes as seen in *Figure 6B* (A). Regression plots for each intervention. Weeks of age (n = 5 per group) (B), weeks of exercise (n = 7) (C), a single dose of β3 adrenergic agonist CL316,243 in mice housed at 30°C (n = 10) (D), and temperature (n = 9) (E).

activity, photoperiod, diet and acclimation were smaller contributors. In the IMPC dataset, temperature and mass were the largest contributors to variation. The lack of accurate temperature information reduced the utility of the IMPC data at many of the sites where even reporting average daily temperature would be beneficial. Other details including the type of indirect calorimeter or body composition method (e.g. DEXA or MRI) would also help interpretation. We recommend that in future publications, the minimal necessary information to compare and interpret metabolic studies (*Table 1*) should be included.

## Sources of variation

At the outset of this project, we envisioned determining the range of metabolic rates observed in WT mice on a LFD or HFD. We predicted a high level of concordance between sites due to the experience, strong reputation for excellence, and know-how in metabolic phenotyping at the participating sites. The MMPC and IMPC experiments were both expressly designed to minimize variation between sites by standardizing protocols, and by studying age and diet-matched animals. Despite these safeguards, we saw a large variance in metabolic rate in each experiment (*Figures 1* and *3*). For the MMPC, these small-scale studies (n = 6–8 mice per group) are typical and can be instructive. In this case, the response of these mice to HFD was highly variable with mice at one site gaining three times more fat mass than mice at another site. Despite the large differences in phenotype, these four MMPC sites produced high-quality data on body mass, locomotor activity, photoperiod, diet, and acclimation which accounted for 80% of the variation. There are no right and wrong phenotypes, only the phenotypic differences in institutional responses, which we have quantified but have yet to fully elucidate. Here temperature and body composition provide the greatest sources of variation.

The IMPC data afford the largest evaluation of non-genetic and genetic sources of variation yet described. This dataset is both impressively large and yet still narrowly focused on a single age and genetic background, on a standard chow diet. Expanding the types of experiments, strains, and covariates beyond those studied by the MMPC and IMPC will be essential for greater understanding. Our analysis helps to quantify the relative contributions of well-known environmental variables including mass, temperature, activity, sex, and season. This was further exemplified both by the comparison of 9358 WT mice (*Figure 3B*) and by comparing the same KO strains at up to nine different institutions (*Figure 5*). We observe phenotypic differences for specific KO strains at some locations, but not at others. However, for most strains we observe good agreement. Understanding the drivers of this variation can help increase reproducibility. Pooling data from multiple sites can also help to reach consensus. The variation or differences among sites is one component of the reproducibility crisis (*Drucker, 2016*) and understanding institutional variability is a key to increasing reproducibility of metabolic data.

**Table 1.** Essential information to report on indirect calorimetry studies.

| |
| --- |
| Location |
| Body mass |
| Body composition |
| Age |
| Accurate ambient temperature |
| Locomotor activity |
| Sex |
| Diet name and composition |
| Strain/Genotype |
| Intervention details |
| Calorimeter model/parameters |
| Body composition method |

## Our current understanding of the critical covariates is as follows

### Sex

We found similar variability of WT female mice compared to male mice in the IMPC experiment. Indeed, we were surprised at the small contribution of sex to metabolic rate. The differences between female and male mice were largely due to the smaller body mass in the former, rather than sex per se. These effects would be predicted to become more pronounced on HFD, as body weights of male and female mice would further diverge and the metabolic effects of altered estrogen and androgen levels may become more prominent.

Body mass: The single most consistent result across the two experimental datasets is the positive correlation between body mass and EE. With sufficient sample size and variation in mass range, this positive relationship can be observed and is one indication of a successful experiment (*Tschöp et al., 2012*).

### Body composition

Typically, inclusion of body composition can improve the quality of regression plots, especially as fat mass accumulates with aging or obesity (*Kaiyala et al., 2010*). In the IMPC experiment, mice were maintained on standard chow diets and had small quantities of body fat. Using body composition data rather than whole body mass as a covariate can help to decrease the variation between groups. It has been proposed that lean mass + 20% fat mass might be used as the covariate for metabolic analysis (*Even and Nadkarni, 2012*), a calculation which has seen limited adoption (*Coyne et al., 2019*; *Ivry Del Moral et al., 2016*; *Dorfman et al., 2017*; *Montgomery et al., 2013*; *Piattini et al., 2019*; *Schöttl et al., 2015*; *Seyfarth et al., 2015*; *Zhao et al., 2016*). We find the contribution of fat mass to whole body EE to be highly variable and dependent on the particular features of the study group. In our study, lean mass + 20% fat mass was not an appropriate mass covariate, and we believe it is unlikely to be a generalizable solution. Furthermore, using lean mass as a covariate in studies of obese mice fails to capture the metabolic contribution imparted by the adipose tissue; an approach that should be treated cautiously (*Tschöp et al., 2012*; *Figure 3—figure supplement 1*).

### Temperature

Adding the *bona fide* experimental temperature improved our ability to account for the variation in metabolic rate at the three sites where this information was available. The powerful role of temperature to affect EE is well established (*Abreu-Vieira et al., 2015*; *Mount and Willmott, 1967*). It therefore stands that providing accurate recorded temperatures would be an easy and effective way to improve the utility of indirect calorimetry results in the IMPC and elsewhere.

### Locomotor activity

Most indirect calorimeters developed in the past 10 years will include the ability to record locomotor activity either by incorporating a calculated distance traveled by counting infrared beam breaks or by tracking a device implanted into the animal. A surprising lack of standardization makes direct comparisons between instruments challenging. However, locomotor activity adds a small but significant contribution to variation in EE and these values should be incorporated whenever possible.

### Acclimation

Acclimation to the indirect calorimetry cages had only a modest effect in the MMPC dataset. Prior to acclimation, mice tended to eat more food than after acclimation. Consequently, for mice on a high-carbohydrate low-fat diet, RER tended to be higher. An acclimation period was not included in a majority of IMPC sites; data were recorded for only 21 hr. The analysis of the MMPC experiments suggest that this had only a small effect on the results presented. However, the rapid adaptation time may be due to the young age of these animals and other experimental strains or conditions may greatly affect acclimation time.

### Sample size

Relatively large numbers of subjects are required to find statistically significant differences between groups using ANCOVA, especially with small variance of body weight or metabolic parameters

(*Arch et al., 2006*; *Even and Nadkarni, 2012*; *Fernández-Verdejo et al., 2019*; *Tschöp et al., 2012*). This point is well illustrated by the large effect of LFD and HFD on mice in the MMPC at UC Davis to promote a change in fat mass from 5 to 14 g in 11 weeks. Yet, despite the obvious phenotypic differences, neither EE nor energy intake was significantly different by ANCOVA in these WT mice with eight animals per group, suggesting that these studies were underpowered. Together, recording and applying these covariates can maximize experimental reproducibility.

## Genetics

The standardization of the IMPC pipeline is an unparalleled resource for re-assessing phenotypic effects of genes with known effects on energy balance. In the scope of performing standardized phenotyping on knockout strains of all mouse gene, several well-characterized genetic experiments have been reproduced by the IMPC consortium members. These genes include those shown to regulate food intake: *Cpe*, *Mrap2*, and *Gdf15*; metabolic rate and thermogenesis: *Ucp1*, *PPARg*, *Gh*, *Gatm*, and *Fgfr4*; Skin barrier function: *Acer1*, and intestinal absorption *Pcsk1*.

## Human GWAS

Our analysis of obesity-related GWAS results specifically focused on strains which may impact EE. Genetic variants identified with these methods are often outside of protein-coding sequences and may affect positive or negative gene regulation (*Buniello et al., 2019*). It is therefore of interest that our analysis predicts KO strains which both increase and decrease metabolic rate. One of the biggest advantages conferred by a repository of metabolic data will be the opportunity for researchers to investigate phenotypes of specific alleles or interventions. In the publication describing the metabolic data from the IMPC, the authors chose a ratio-based reporting system based on a residuals from multiple linear regression (*Rozman et al., 2018*). We have reported the residual as well as standard deviation from the mean. Both approaches are similar, but we find reporting the average mass and deviations of the KO strain can add a dimension of context to the results. The strains we identified have not been well characterized with regard to their roles in metabolic pathways, suggesting follow-up studies are warranted.

Strains with lower metabolic rate which might contribute to development of obesity include *Chst8*, *Pax5*, *Pald1, and Tfap2b*. *Chst8*, carbohydrate sulfotransferase 8 mediates sulfation of the carbohydrate structures of the sex hormone LH; deletion of *Chst8* produces mice with elevated LH levels. *Chst8* is also expressed in the hypothalamus and pituitary and is predicted to modify proopiomelanocortin (POMC), the precursor to several key hormones involved in the control of stress and body weight regulation (*Mi et al., 2008*). High-throughput IMPC analysis revealed decreased locomotor activity in *Chst8*$^{-/-}$ mice. *Pald1*, phosphatase domain containing paladin 1, is a regulator of angiogenesis and vascular function. Female *Pald1*$^{-/-}$ exhibit an emphysema-like lung disorder (*Egaña et al., 2017*). However, cell culture studies have shown *Pald1* to have a role in insulin signaling by negatively regulating insulin receptor expression and phosphorylation (*Huang et al., 2009*). *Pax5*, paired box 5, is a transcription factor controlling B cell development (*Nutt et al., 1999*). IMPC analysis confirmed lower leukocyte counts and also found increased cardiac ejection fraction suggestive of cardiopulmonary phenotypes in the *Pax5*$^{+/-}$ mice. *Tfap2b*, transcription factor AP-2 beta, KO mice exhibit a form of patent ductus arteriosus (*Satoda et al., 2000*). However *Tfap2b* heterozygous mice are largely unaffected (*Zhao et al., 2011*), findings confirmed in IMPC analysis.

Strains with increased metabolic rate include *Pepd*, *Klf12*, and *Pacs1. Pepd*, peptidase D (or prolidase) is an enzyme which cleaves dipeptides including collagen. Patients harboring mutations in *PEPD* have skin ulcers (*Sheffield et al., 1977*). *Pepd* KO mice reveal an important role for this protein in bone and hematopoetic development. These mice have defects including alterations in bone and immune development (*Besio et al., 2015*). The *Pepd* gene affects coat pigmentation, suggesting an interaction with melanocortin signaling pathways (*Cota et al., 2008*). IMPC analysis reveals also numerous behavioral and developmental defects. *Klf12*, kruppel like factor 12 is a transcriptional repressor linked by GWAS to EE (*Jiang et al., 2018*). *Klf12* is necessary for NK cell proliferation in mice (*Lam et al., 2019*) and IMPC studies reveal lower levels of locomotor activity. *Pacs1,* phosphofurin acidic cluster sorting protein 1, is a protein with a putative role in the localization of trans-Golgi network membrane proteins and has been shown to be important for vesicular trafficking of POMC

into dense secretary granules in neuroendocrine cells, a pathway essential to its downstream cleavage into its active peptide hormones. *Pacs1* may also affect the function of cilia, a structures that when altered are associated with development of obesity (*Schermer et al., 2005*). While Pacs1 homozygous deletion mice were embryonic lethal, heterozygote mice revealed no suggested pathology under unchallenged conditions.

In summary, these seven genes may affect EE through roles in POMC or melanocortin signaling, skin barrier function, lung defects, hematopoiesis, and bone formation. Follow-up studies will help to clarify the role of these proteins in mouse biology. Greater understanding in mice would provide testable hypotheses for assignment of causal human genetic variants to clinical phenotypes.

## Limitations

Several key limitations of the current study include the currently accepted methodology for analysis which requires reduction of hundreds of data measurements collected for each animal down to a single value for ANCOVA or multiple linear regression-based statistical analysis. The IMPC data provides one data point per metabolic variable (e.g. oxygen consumption or EE) per animal. Further research on time-series could potentially extract meaningful information from these discarded values. One possible source of unexplained variation between sites is the different models of indirect calorimetry systems used. This study was unable to directly compare the efficacy of different indirect calorimeter manufacturers due to the large institutional differences in EE even between sites using the same instrumentation. However, a direct comparison of results from mice with two different systems within the same room has recently been reported (*Soto et al., 2019*). Different calorimeter parameters and calibrations, such as how many chambers feed into a single gas analyzer, and humidity, pressure and flow rate of gases, can influence results and should be included when reporting data (*Table 1*). It is not common practice for investigators to calibrate the whole mouse calorimetry system (calorimetry chambers through the gas analyzers). However, this whole system calibration can be accomplished by introducing a standard gas of known composition into each chamber at a controlled flow rate and measuring recovery of carbon dioxide and oxygen. Appropriate software modules can be used to ensure the accuracy of calibrations and data accuracy. This type of calibration system should be encouraged to facilitate comparisons of indirect calorimetry apparatus. The process of observing mice in an indirect calorimeter may affect their behavior and fail to accurately reflect normal food intake, locomotion, and EE. The energy balance calculated for the MMPC experiment is inconsistent with the body weight gain over the 11-week period on HFD. This finding emphasizes the difficulty in accurately determining energy balance over a few days and may be better captured by longer measurements. It is also possible that the calorimetry environment altered energy balance compared to home cage conditions. It can be challenging to accurately measure energy intake and systems designed to minimize spillage could also impact food intake by making it more difficult for mice to eat. Observing serial measurements of body weight can help to mitigate interpretation of experiments where data unexpectedly reflects weight loss (as in *Figure 1*). Both in-depth phenotyping and large-scale population-based phenotyping reveal the large effect size of institutional variation. Two possible but untested sources of variation in these datasets include changes in gut microbiota and/or epigenetic changes induced by different environmental triggers. Mice have specific microbial flora which change over time and affect energy balance (*Turnbaugh et al., 2006*). The microbiome is likely to be an institutionally local phenomenon which can affect metabolic rates; indeed studies linking the dependence on thermogenesis with the microbiome have reported different results in experiments performed in Boston, New York City, Geneva and Beijing (*Chevalier et al., 2015*; *Krisko et al., 2020*; *Li et al., 2019*). Knowledge of microbial populations which correlate with EE may help to refine metabolic studies in the future. Attempts to rescue phenotypic variation by microbiome reconstitution may prove fruitful. Similarly, it may soon be possible to interrogate the set of epigenetic modifications to genes controlling metabolic rate and help to predict or modify metabolic rate accordingly.

## Conclusions

There is a surprisingly large degree of phenotypic variation observed in mouse energy expenditure. The two experimental paradigms analyzed were deliberately designed to create consistent, reproducible results. Contrary to our predictions, there were still unaccounted-for differences among

institutions either in the response of WT animals to differentially gain body weight or with intrinsic differences in energy expenditure due to uncontrolled factors such as ambient temperature. This variation was sufficiently large to prevent consistent phenotypic conclusions caused by the same genetic interventions. The experimental location variability strongly affected results in both WT and genetically modified strains. These findings suggest that reporting experimental conditions including body composition, accurate temperature, and activity should be essential to reproducing and comparing calorimetry data, and can explain the majority of institutional differences. Identifying the as-yet unknown sources of experimental variability among sites should also become a priority to foster consistency and reproducibility in experimental results among multiple institutions. However, within any one institution, we can trust that results obtained comparing littermate controls using ANCOVA for analysis are experimentally valid for those conditions. Yet investigators dedicated the reproducibility of their experimental model at other sites will need to report the essential information for interpretation of indirect calorimetry studies (*Table 1*). Using this model, individual labs or centers may leverage the big data approach to understanding phenotypes by meticulously creating datasets of non-littermate control mice (as in the IMPC data, *Figure 6*). Once validated, these data can be used to regress against smaller cohorts of experimental mice, as well as controls to validate the approach in each instance. This strategy suggests that indirect calorimetry can still be a useful tool to understand metabolic phenotypes.

## Materials and methods

### MMPC experimental description

Data from the NIH-funded MMPCs (RRID:SCR_008997) represent longitudinal measurements at 4 sites located at University of California, Davis (UC Davis), University of Massachusetts (UMass), Yale University (Yale) and Vanderbilt University (Vanderbilt). Indirect calorimetry measurements were recorded for each mouse for at least 4 days, allowing for analysis of pre-acclimation (the first 18 hr) and post-acclimation (18–96 hr). Unless otherwise noted, our analyses use the post-acclimation data. The 4 geographically distinct MMPC sites used calorimeters from 3 different manufacturers (Columbus Instruments at UC Davis, California and Yale, Connecticut; TSE Systems at UMass, Massachusetts; Sable Systems International at Vanderbilt, Tennessee). Genetically identical C57BL/6J male mice (n = 60, 6–7 weeks of age) originating from the same room in the Jackson Laboratory facility were divided into four groups, and each group was shipped to one site. On arrival, mice were maintained on LFD for one week and then randomized into two groups for all further experiments (n = 6–8 mice per diet per site). To control for the transition from group to single housing, all animals remained singly housed for the duration of the study. Indirect calorimetry was performed on all mice at each site (week 0) after which one group was given HFD for the next 12 weeks, while the other was maintained on LFD. All mice were returned to the calorimeters at 4 and 11 weeks post-diet randomization. Mouse diets were purchased as a uniform lot from a singular production stream for all sites from Research Diets (LFD: D12450B, 10% energy derived from fat, 15.69 kJ/g; HFD: D12452, 60% energy derived from fat, 21.92 kJ/g) and were delivered to each site simultaneously. Energy intake was calculated by taking the product of food intake in grams with the energy density of the diet in kJ. UC Davis used a PIXIMus DEXA scanner under anesthesia for body composition measurements. Other sites used an NMR-based Bruker minispec without anesthesia. Reported room temperatures for the UC Davis, UMass, Yale, and Vanderbilt MMPC sites were 22.0, 21.1, 21.5°C and 22.5°C respectively. For indirect calorimetry, the Vanderbilt site used a temperature-controlled environment of 24°C. Beam breaks reported from Columbus Instruments, Sable Systems International, and TSE Systems correspond to different distances. Accordingly, locomotor activity was calculated as beam breaks as a percent of the global maximum per site.

### IMPC experimental description

IMPC (RRID:SCR_006158) data were collected as described (*Rozman et al., 2018*). In version 10.0 of this dataset, the 24 hr averaged metabolic data from more than 30,000 mice is publicly available. The IMPC data uses highly similar indirect calorimetry protocols across 10 sites in 8 countries. All indirect calorimetry data were collected from 11-week-old mice, except for body composition data. Lean and fat masses corresponding to week 11 were estimated for each mouse based on body

composition percentages collected at week 14. The calorie content and macronutrient compositions of the chow diets used were provided when requested. The IMPC 10.0 sites uniformly provide a single average value reported for oxygen consumption, carbon dioxide release, RER, and EE, per animal although additional hourly data is available through their API. Data were not consistently present at all sites for both males and females, and sites did not uniformly report locomotor activity, precise housing temperatures, or food intake data. Similarly, acclimation was performed only at a subset of sites and pre-acclimation data was not readily available. On initial visualization, the data deposited showed several abnormalities. These included one site where larger animals paradoxically consumed less oxygen than smaller animals, demonstrating an inverse mass effect (*Figure 3—figure supplement 3A*). When contacted this site quickly identified and corrected a persistent error in their data analysis pipeline which will be rectified in a future IMPC data release. A different site reported erroneous food intake data with each animal consuming exactly 0.05 g of food per hour regardless of body mass or genotype. This site also provided an updated dataset with corrected food intake values (*Figure 3—figure supplement 3B*). We also observed differences in methods to calculate EE from indirect calorimetry data. When calculating EE using the Weir formula (*Weir, 1949*) we find two sites which produced different values, likely due to implementation of other formulae including the Lusk equation (*Lusk, 1993*; *Figure 3—figure supplement 3C*). Lastly, we identified multiple sites providing data from calorimetry experiments shorter than the reported 21 hr minimum duration. At one site, the full run data was erroneously excluded from the IMPC database, and once contacted the complete data was provided for use in this analysis. All other experiments with durations shorter than 18 hr were excluded from our analysis (*Figure 3—figure supplement 3D*). To improve consistency, we re-calculated data for all mice using the Weir equation. Erroneously keyed data were corrected as described in *Figure 3—figure supplement 3*. One strain was excluded due to highly variable values within the genotype in a subset of mice, *Fbxl19*. The acronyms describing the IMPC sites have changed in some instances. For simplicity, the following sites are indicated by the country in which they are located. For countries with two sites, the state or county is indicated (*Table 2*).

## Other data sources for indirect calorimetry measurements

All data were collected with Columbus Instruments CLAMS indirect calorimeters in male mice. Data demonstrating the effect of age in chow-fed (Teklad 2018) C57BL/6J mice were graciously contributed by *Houtkooper et al. (2011)*. EE values during voluntary wheel running exercise in 6-month-old chow-fed (Teklad 7022) C57BL/6 male mice at 22°C are as reported in the supplemental materials from *O'Neal et al. (2017)*. Studies of the effect of β3-adrenergic stimulation and of temperature change were performed in a temperature-controlled chamber enclosing the indirect calorimeter. Both studies were conducted in Boston MA. For β3 stimulation, 6.5-month-old chow-fed (LabDiet 5053) C57BL/6J global *Cdkal1*$^{-/-}$ or WT littermate male mice were maintained at 30°C for 24 hr. Mice received an IP administration of 1 mg/kg CL, 5 hr into the light photoperiod. The mean EE over the 3 hr post-injection period was used in this analysis. No significant differences in EE were observed

**Table 2.** IMPC sites and abbreviations.

| Name in this study | Location | IMPC abbreviation |
|---|---|---|
| California, USA | University of California, Davis | UC Davis |
| Cambridgeshire, UK | Wellcome Trust Sanger Institute | WTSI |
| Canada | Toronto Centre for Phenogenomics, The Centre for Phenogenomics | TCP |
| China | Model Animal Research Center of Nanjing University | MARC |
| France | PHENOMIN-Institut Clinique de la Souris | ICS |
| Germany | Helmholtz-Zentrum Muenchen | HMGU |
| Japan | RIKEN BioResource Research Center | RBRC |
| Korea | Korea Mouse Phenotype Consortium | KMPC |
| Oxfordshire, UK | Medical Research Council, Harwell | MRC Harwell |
| Texas, USA | Baylor College of Medicine | BCM |

between genotypes; accordingly, data for all mice were pooled. EE measurements of $Cdkal1^{-/-}$ or WT littermate control male mice maintained at 23°C were used in *Figure 5*. The study of the effect of temperature on EE was performed on 10-week-old chow-fed (LabDiet 5053) C57BL/6J male mice with an incremental decrease in temperature. Mice were maintained at 30°C for 24 hr intervals between each 24 hr temperature challenge ranging from 28°C to 6°C.

### *CalR* analysis

*CalR* analysis (RRID:SCR_015849) was performed on the MMPC dataset (*Figure 1—figure supplement 1*) as described (*Mina et al., 2018*). The 'remove outliers' feature was enabled to exclude from analysis data that were recorded during momentary cage opening.

### Animal experiments

Previously unpublished studies were performed in strict accordance with the recommendations in the Guide for the Care and Use of Laboratory Animals of the National Institutes of Health. All the animals were handled according to approved institutional animal care and use committee (IACUC) at the site where they were performed.

### Statistical methods

Data input, cleaning, visualization, and statistical analysis were performed in the R programming language (*R Development Core Team, 2019*). The relative contribution of covariates were calculated with the relaimpo package (*Grömping, 2006*) The multiple linear regression model and model to quantify the explained variance for the MMPC experiment after 11 weeks on diet included body composition, locomotor activity, photoperiod, diet, and acclimation. These models were not improved by the addition of equipment manufacturer or temperature as each site reported a single, unique temperature, and only two sites used a similar manufacturer of indirect calorimeter. The IMPC models included body composition, locomotor activity, ambient temperature, sex and season. These models were not improved with addition of equipment manufacturer. The q-q plot was performed with a geom_qq using EE data from WT mice at the seven sites examining both sexes (*Wickham, 2009*). All plots were produced with ggplot2 or *CalR* (*Mina et al., 2018*; *Wickham, 2009*). Unless otherwise specified, all reported data are based on the daily EE (DEE), 24 hr averaged EE values. Experiments were not performed at thermoneutrality, eliminating the possibility of determining basal EE or resting metabolic rate at thermoneutrality (*Even and Nadkarni, 2012*; *Meyer et al., 2015*).

## Acknowledgements

Financial support for this work was provided by the NIDDK Mouse Metabolic Phenotyping Centers (National MMPC, RRID:SCR_008997, www.mmpc.org) under the MICROMouse Program, grant U24-DK076169 (ASB) and R01DK107717 (ASB), and Swiss National Science Foundation Postdoc Mobility grant to DR. We acknowledge Drs. K C Kent Lloyd (National Institute of Diabetes and Digestive and Kidney Diseases (NIDDK) U24-DK092993, Gerald I Shulman (NIDDK U24-DK059635), and Henri Brunengraber (NIDDK U24-DK076174), David H Wasserman (U24-DK059637), Patrick Tso (NIDDK U24-DK059630), Jason Kim (U24-DK093000) and Richard McIndoe (U24-DK076169) for their support of this work. We thank Christopher Jacobs and Rachael Ivison for fruitful discussions on bioinformatic approaches, Jeff and Terry Flier for critical reading of the manuscript.

MMPC Energy Balance Working Group Members include: Owen P McGuinness, Louise Lantier, Jon J Ramsey, Collen Croniger, Randall H Friedline, Sean Adams, Heni Brunengraber, Michael Jurczak, Li Kang, Jason K Kim, Kent Lloyd, Richard McIndoe, Silvana Obici, Jerry Shulman, Craig Warden, Thomas Gettys, David Wasserman, Trina A Knotts, Karl Kaiyala.

## Additional information

### Funding

| Funder | Grant reference number | Author |
|---|---|---|
| National Institutes of Health | R01-DK107717 | Alexander S Banks |
| National Institutes of Health | U24-DK092993 | Jon J Ramsey |
| National Institutes of Health | U24-DK059635 | Michael J Jurczak |
| National Institutes of Health | U24-DK076174 | Owen P McGuinness |
| National Institutes of Health | U24-DK059637 | Owen P McGuinness |
| National Institutes of Health | U24-DK059630 | Owen P McGuinness |
| National Institutes of Health | U24-DK093000 | Jason K Kim |
| National Institutes of Health | U24-DK076169 | Alexander Banks |
| Swiss National Science Foundation | Postdoc Mobility Grant | Deepti Ramachandran |

The funders had no role in study design, data collection and interpretation, or the decision to submit the work for publication.

### Author contributions

June K Corrigan, Conceptualization, Data curation, Software, Formal analysis, Investigation, Visualization; Deepti Ramachandran, Writing - original draft, Writing - review and editing; Yuchen He, Colin J Palmer, Michael J Jurczak, Randall H Friedline, Jason K Kim, Jon J Ramsey, Investigation; Rui Chen, Data curation; Bingshan Li, Formal analysis; Louise Lantier, Conceptualization, Investigation; Owen P McGuinness, Conceptualization, Funding acquisition; Alexander S Banks, Conceptualization, Data curation, Formal analysis, Supervision, Funding acquisition, Visualization

### Author ORCIDs

June K Corrigan (ID) https://orcid.org/0000-0002-7514-0177
Deepti Ramachandran (ID) http://orcid.org/0000-0003-1113-1295
Louise Lantier (ID) http://orcid.org/0000-0002-6620-4976
Owen P McGuinness (ID) http://orcid.org/0000-0002-1778-3203
Alexander S Banks (ID) https://orcid.org/0000-0003-1787-6925

### Ethics

Animal experimentation: These studies were performed in strict accordance with the recommendations in the Guide for the Care and Use of Laboratory Animals of the National Institutes of Health. All of the animals were handled according to approved institutional animal care and use committee (IACUC) at the site where they were performed.

### Decision letter and Author response

Decision letter https://doi.org/10.7554/eLife.53560.sa1
Author response https://doi.org/10.7554/eLife.53560.sa2

## Additional files

### Supplementary files
• Transparent reporting form

### Data availability

All data and code can be found at https://github.com/banks-lab/Cal-Repository copy archived at https://github.com/elifesciences-publications/Cal-Repository. Repository data includes complete

indirect calorimetry data for MMPC experiments including CalR files for 4 sites at 0, 4, and 11 week trials, our MMPC analysis database, corrected IMPC database, and additional data for Figures 5 and 7. The R code to reproduce all figures is also included.

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
