## [Decision Letter]

Thank you for submitting your article "A big-data approach to understanding metabolic rate and response to obesity in laboratory mice" for consideration by *eLife*. Your article has been reviewed by three peer reviewers, one of whom is a member of our Board of Reviewing Editors, and the evaluation has been overseen by Mark McCarthy as the Senior Editor. The following individual involved in review of your submission has agreed to reveal their identity: John R Speakman (Reviewer #3).

The reviewers have discussed the reviews with one another and the Reviewing Editor has drafted this decision to help you prepare a revised submission.

Summary:

Corrigan and colleagues offer a large-scale, multi-site analysis of indirect calorimetry as a tool for determining metabolic rate. The authors should be applauded for trying to provide systematic and rational guidance for interpreting this type of specific data sets. However, several issues need to be addressed. Despite the impressive attention to detail in the experimental approach (age-matched mice from the same room at Jackson, diets from the same lot and shipment, analysis of both MMPC and IMPC datasets), the final conclusion appears to be more of a cautionary tale against the over-reliance and over-interpretation of indirect calorimetry, rather than a useful set of guidelines that can be used by all investigators. The authors need to deal with a potential artifact of the way the analysis regarding fundamental rules for multiple regression analysis.

Essential revisions:

A big picture question not really addressed are these institutional variations addressable or does this study call into question the usefulness of indirect calorimetry to provide meaningful insights into metabolic rate?

The model represented in Figure 1F depends on a multiple regression analysis but a fundamental assumption of such an analysis is that the independent variables are independent – by definition. In this case they are not because there are institutional differences in lean and fat mass as shown in Figure 1E. This would tend then to diminish the variance attributable to the traits that differ within site. Indeed lean mass and mass here account for only about 30% of the variation in the EE while other studies indicate much larger explained variation by these two factors (see eg Kaiyala's work). I think it would therefore be better to first omit institution from the analysis – find the effects of lean mass, fat mass, locomotor activity etc and then ask if institution explains any of the residual variation once these biological factors are accounted for.

The most striking finding is how much inter-institutional variation exists between identical experiments. However, there are no suggestions as to how this can be overcome. Is there some standard that all systems can be calibrated to?

It is nonetheless of considerable value as it nicely compares the impact of covariates such as sex, locomotion, body weight, temperature or body weight and body composition on EE variance for a large body of data. Of particularly interest are further the EE data of KO mice that had been linked to obesity in GWAS analyses. First, rather preliminary, assessments indeed suggest that these genes are involved in the regulation of energy homeostasis. This finding would warrant a more comprehensive display of available IMPC data. Furthermore, did the authors try to compute individual factors for the different institutions that would allow to normalize the data between sites? Would such an adjustment normalize the residuals for some of the KOs in Figure 6D,F?

The dominant institutional effect is actually not proven because of the confounding collinearity in the predictors. Institution clearly is not independent of the other factors so the way it is treated in the analysis is key. I strongly advise fitting the biology to the data and then afterwards asking if institution explains any residual variance. That will give a much more representative picture of the institutional effect.

Were environmental measures such as humidity, luminance measured in each room? Were the time of the day when experiments were conducted similar?

The fact that mice at UMass consistently show an RER significantly above 1 during the dark cycle (Figure 1—figure supplement 1E), suggests that this is needed. Is there a way to present raw data, independent of the analysis algorithms of each system? Are they suggestive of an unappreciated biological complexity in that mouse model, or a systemic flaw in that institution's set up?

The resource fails to report how large difference in body weight/composition may affect the data (e.g. comparing a very obese KO model to a lean WT). Will the same ANCOVA analysis be sufficiently robust if animals differ in X % of body weight or lean mass? What should be the limit?

The authors suggest changes in gut microbiota and/or epigenetic changes induced by different environmental triggers. Providing such data would likely be a decisive advance for our field. However, as it now stands it may be better not to specify the factors but just say “site specific factors”.

The analysis here is interesting and sobering for those of us trying to discern effects of genotype on metabolism but there is still the collinearity issue in the predictors of the linear multiple regression (subsection “Variability in KO phenotypes”).

---

## [Author Response]

Essential revisions:A big picture question not really addressed are these institutional variations addressable or does this study call into question the usefulness of indirect calorimetry to provide meaningful insights into metabolic rate?The model represented in Figure 1F depends on a multiple regression analysis but a fundamental assumption of such an analysis is that the independent variables are independent – by definition. In this case they are not because there are institutional differences in lean and fat mass as shown in Figure 1E. This would tend then to diminish the variance attributable to the traits that differ within site. Indeed lean mass and mass here account for only about 30% of the variation in the EE while other studies indicate much larger explained variation by these two factors (see eg Kaiyala's work). I think it would therefore be better to first omit institution from the analysis – find the effects of lean mass, fat mass, locomotor activity etc and then ask if institution explains any of the residual variation once these biological factors are accounted for.

The reviewers have hit on a key point that we overlooked during our initial submission. We agree wholeheartedly that the true differences in the MMPC dataset are the different responses observed in body weight at the different institutions and that these differences in body weight are driving the institutional differences in energy expenditure. The suggested approach is more appropriate for examining the true source of the institutional variation. We have removed “institution” from the MMPC and IMPC models used in Figures 1, 3, and 4. We thank the reviewers for this excellent point.

Indeed, the original MMPC model indicated that institution, lean mass and fat mass are more correlated with other predictors than with EE. When removing institution as a factor from the model, the R2 value is 72.41%. Only 16.3% of the residual variance in the model is further explained by institution. Variations in lean and fat mass now account for ~50% of EE variation, more consistent with the results of Kaiyala, as suggested.

For the IMPC experiment, despite the smaller mass range of these chow-fed animals, we also detect high levels of multicollinearity in the model. We find that institution and ambient room temperature are more correlated with other predictors than with EE. The collinearity with room temperature is likely due to the inaccurate reporting at multiple sites. Removing institution decreases the R2 value to 38.82%, with temperature now accounting for 23% of EE variance. 9.6% of the residual variance in the model is explained by institution.

In the comparison of the 3 IMPC sites that report accurate temperatures (Figure 4), the R2 value with institution removed from the model is 67.20%. Here, a mere 0.078% of residual variation is explained by institution, emphasizing that body composition and ambient temperature are critical covariates in determining EE and thus can be major influences in variability among institutions. We acknowledge and appreciate the reviewer’s correct suggestion as to the actual source of the institutional variation in these models.

The most striking finding is how much inter-institutional variation exists between identical experiments. However, there are no suggestions as to how this can be overcome. Is there some standard that all systems can be calibrated to?

Because of the design of both the MMPC and IMPC experiments, it is impossible for us to directly examine the effects of different calorimetry systems, or whether they were appropriately calibrated. We and others have thought a great deal about how to ensure that calorimetry systems are appropriately calibrated and whether there was some way to do this. The theoretical solution involves both hardware and software. On the hardware side, instruments would first undergo a standard calibration of the indirect calorimeter by the manufacturer’s specifications. To test the calibration, we would then to infuse a precisely known amount of O2 and CO2 into an empty indirect calorimetry cages (e.g. sufficient to mimic an increase of 20 ml/hr over baseline readings). Software which could read files from any of the manufacturer’s file formats could use these reading to adjust the rest of an experimental run appropriately. This test would work for any system and could be performed at any time during an experiment. A hardware system similar to this is implemented at UC Davis. Drs. McGuiness, Lantier and Banks are members of the MMPC reproducibility committee and is working on describing and implementing such a solution more broadly.

It is nonetheless of considerable value as it nicely compares the impact of covariates such as sex, locomotion, body weight, temperature or body weight and body composition on EE variance for a large body of data. Of particularly interest are further the EE data of KO mice that had been linked to obesity in GWAS analyses. First, rather preliminary, assessments indeed suggest that these genes are involved in the regulation of energy homeostasis. This finding would warrant a more comprehensive display of available IMPC data. Furthermore, did the authors try to compute individual factors for the different institutions that would allow to normalize the data between sites? Would such an adjustment normalize the residuals for some of the KOs in Figure 6D,F?

We thank the reviewers for this comment on the value of our analyses. In the original model, we did include institution into the model shown in Figure 6, along with body mass, sex, and ambient temperature. However, given the advice of the reviewers to remove institution from our model due to profound interdependence between institutional and biological factors, it is not appropriate to compute separate coefficients for each institution.

The dominant institutional effect is actually not proven because of the confounding collinearity in the predictors. Institution clearly is not independent of the other factors so the way it is treated in the analysis is key. I strongly advise fitting the biology to the data and then afterwards asking if institution explains any residual variance. That will give a much more representative picture of the institutional effect.

We have since changed this analysis and conclusion – please see earlier response.

Were environmental measures such as humidity, luminance measured in each room? Were the time of the day when experiments were conducted similar?

We thank the reviewers for emphasizing the importance of these factors in conducting metabolic studies. For the MMPC experiments, all runs were recorded over the course of 4 days, with all mice placed in the calorimeters during the light cycle and acclimated for at least 18 hours. Humidity and luminance, along with other environmental measures, are unknown.

For the IMPC experiments, all runs were conducted beginning 5 hours before the start of the dark cycle, and concluded at least 4 hours after the start of the subsequent light cycle. Details about humidity and luminance have been requested but we have not yet received a response about this and other environmental measures.

The fact that mice at UMass consistently show an RER significantly above 1 during the dark cycle (Figure 1—figure supplement 1E), suggests that this is needed. Is there a way to present raw data, independent of the analysis algorithms of each system? Are they suggestive of an unappreciated biological complexity in that mouse model, or a systemic flaw in that institution's set up?

We thank the reviewers for noting these unusually high RER values at Umass for mice on a standard low-fat diet at initial observation (0 wks). UMass uses a TSE system, unlike the other 3 MMPC sites, which may contribute to differences in VO2 and VCO2 measurements, though we have no information on system calibration or flow settings. We have displayed in Author response image 1 the time plots of VO2, VCO2 and RER at UMass in context with the other sites (top row), in isolation (middle) and by individual mice (bottom). This figure demonstrates that the RER observed is not due to an outlier. We would also like to stipulate that no gas exchange measurements were excluded from the UMass group, all of the data provided were within 3 standard deviations for the given photoperiod. These high RER readings may be due to an improperly calibrated system or other institutional variations observed at the Umass site.

The resource fails to report how large difference in body weight/composition may affect the data (e.g. comparing a very obese KO model to a lean WT). Will the same ANCOVA analysis be sufficiently robust if animals differ in X % of body weight or lean mass? What should be the limit?

This is a great point. Multiple prior publications suggested that the ANCOVA was an appropriate way to compare mice with different body weights and Kaiyala et al. specifically addressed differences in body composition. The large IMPC dataset is able to inform the question of when is a difference in body weight too large for appropriate comparisons.

We approached this question with the IMPC data including all male and female WT mice grouped into quartiles (Q1 small, 14.00-20.75 g; Q2 medium, 20.75-27.50 g; Q3 large, 27.50-34.25 g; Q4 largest, 34.25-41.00 g), there is a significant interaction between total mass and EE (a mass effect). We find that there were significant differences in the slope of the relationship between EE and body mass among the four groups. Here the “large” mice have a significantly shallower slope in the relationship between body mass and EE compared to both small and medium mice. This finding was similar to the relationship observed in the MMPC cohort on LFD and HFD (Figure 2A). This is due to the differential contribution of lean and fat mass to EE which are not equally accumulated in mice with greater body weight. This finding suggests that when comparing mice with more than 20% difference in body mass, the body composition data must be added to an ANCOVA model to account for these differences in body composition. Somewhat surprisingly, the largest mice are not significantly different from any of the other groups, which may be attributed to the fact that there are the fewest mice in the group with the largest masses. These data are now included as Figure 3—figure supplement 2. We find no significant differences in EE vs body mass for male and female WT mice grouped into quartiles of lean mass.

The authors suggest changes in gut microbiota and/or epigenetic changes induced by different environmental triggers. Providing such data would likely be a decisive advance for our field. However, as it now stands it may be better not to specify the factors but just say “site specific factors”.

We agree that there is ongoing debate about the role of the microbiome in metabolism. This point is emphasized by the fact that another reviewer specifically suggested we cite a microbiome study, where Li et al., 2019, found that mice lacking gut microbiota had impaired thermogenesis in the cold. However, a recent study by Krisko et al., 2020, found no influence on thermogenesis by microbiome. The authors of this latter study posit that differences in microbiome compositions may be responsible for these different factors. Because there are conflicting findings about the relationship between microbiome and energy expenditure, we feel that it is important to note its possible role in influencing EE.

The analysis here is interesting and sobering for those of us trying to discern effects of genotype on metabolism but there is still the collinearity issue in the predictors of the linear multiple regression (subsection “Variability in KO phenotypes”).

Thanks to the reviewers again for emphasizing the collinearity issue undermining our original analyses of institutional effect. Please see response above.